# microRNA Expression of Renal Proximal Tubular Epithelial Cells and Their Extracellular Vesicles in an Inflammatory Microenvironment In Vitro

**DOI:** 10.3390/ijms241311069

**Published:** 2023-07-04

**Authors:** Patrick C. Baer, Ann-Kathrin Neuhoff, Ralf Schubert

**Affiliations:** 1Department of Internal Medicine 4, Nephrology, University Hospital, Goethe-University, 60596 Frankfurt/M., Germany; 2Division of Allergology, Pneumology and Cystic Fibrosis, Department for Children and Adolescents, University Hospital, Goethe-University, 60596 Frankfurt/M., Germany

**Keywords:** renal tubular cells, epithelial cells, proximal tubule, extracellular vesicles, cytokines, inflammation, inflammatory microenvironment, kidney

## Abstract

Renal proximal tubular epithelial cells (PTCs) are central players during renal inflammation. In response to inflammatory signals, PTCs not only self-express altered mRNAs, microRNAs (miRNAs), proteins, and lipids, but also release altered extracellular vesicles (EVs). These EVs also carry inflammation-specific cargo molecules and are key players in cell–cell-communication. Understanding the precise molecular and cellular mechanisms that lead to inflammation in the kidney is the most important way to identify early targets for the prevention or treatment of acute kidney injury. Therefore, highly purified human PTCs were used as an in vitro model to study the cellular response to an inflammatory microenvironment. A cytokine-induced inflammatory system was established to analyze different miRNA expression in cells and their EVs. In detail, we characterized the altered miR expression of PTCs and their released EVs during induced inflammation and showed that 12 miRNAs were significantly regulated in PTCs (6 upregulated and 6 downregulated) and 9 miRNAs in EVs (8 upregulated and 1 downregulated). We also showed that only three of the miRNAs were found to overlap between cells and EVs. As shown by the KEGG pathway analysis, these three miRNAs (miR-146a-5p, miR-147b, and miR-155-5p) are functionally involved in the regulation of the Toll-like receptor signaling pathway and significantly correlated with the inflammatory mediators IL6 and ICAM1 released by stimulated PTCs. Especially with regard to a possible clinical use of miRs as new biomarkers, an accurate characterization of the miR expression altered during inflammatory processes is of enormous importance.

## 1. Introduction

Renal proximal tubular epithelial cells (PTCs) are not only an integral part of the nephron, which is responsible for the reabsorption of water, electrolytes, and other substances from the filtrate produced by the glomerulus, but they are also central players during renal inflammation and mediate the response to inflammation during kidney diseases [1]. Inflammation in the proximal tubular segment is often associated with an immune response to injury or infection [2]. The inflammatory process involves the activation of immune cells, the release of pro-inflammatory molecules, the recruitment of inflammatory cells to the affected area, and tissue damage. This inflammation can lead to various renal diseases, including acute and chronic renal injury, tubulointerstitial nephritis, and finally, fibrosis [3,4].

During the inflammatory process, renal tubular epithelial cells are exposed to various stimuli, such as cytokines, chemokines, and damage-associated molecular patterns. In response to these inflammatory signals, PTC release extracellular vesicles (EVs) that carry inflammation-specific cargo molecules [5]. These EVs are than taken up by neighboring cells or transported to distant sites, where they can influence the immune response and contribute to the progression or resolution of inflammation. EVs are small membrane-bound particles released by nearly all cell types, including renal tubular epithelial cells, and they play crucial roles in cell-to-cell communication. There are different types of known EVs, leading to their classification according to their sub-cellular origin, release pathways, and size [6]. On the one hand, there are EVs released by budding from the cells’ plasma membrane, known as microvesicles, ectosomes, or microparticles. On the other hand, there are EVs which are generated inside multivesicular bodies and are secreted by the fusion of the body with the plasma membrane, known as exosomes. Both subtypes are mainly involved in intercellular communication in health and disease. Besides these types, there are also apoptotic bodies belonging to the class of EVS, which, however, have a different task compared to the other subgroups. Apoptotic vesicles are released from the plasma membrane during programmed cell death and therefore transport substances from dying cells [7].

All EVs exhibit some diversity, particularly in terms of their surface molecules, which allows them to be targeted to recipient cells. It has been shown that when EVs attach to a target cell, they trigger signaling pathways via receptor–ligand interactions or internalization via endocytosis or phagocytosis or even by fusion with the membrane of the target cell [8]. It has also been shown that EVs from parent cells can interact in a specific manner with recipient cells, e.g., in the renal nephron, and it has further been suggested that a specific signaling pathway exists from proximal to distal tubular epithelial cells [9]. The signaling pathways and communication via EVs are mostly accomplished by nucleic acids, lipids, proteins, and carbohydrates included in the EVs. By delivering their content into the cytosol of a target cell, EVs can modify the physiological state of the recipient cell [8]. In particular, small RNA molecules, so-called microRNAs (miRNAs), were proven to play a central role as epigenetic regulators in this context, since they regulate gene expression via post-transcriptional mechanisms. Therefore, they are touted as the next generation of biomarkers for different human diseases and biological states [9]. They were, inter alia, shown to prevent the uncontrollable progression of inflammation but were also shown to be frequently dysregulated in the development of renal fibrosis and other diseases [10,11]. Since free RNA included in serum or medium is relatively instable, EVs could serve as a protector of RNA from degradation and therefore contribute to the status of a potential biomarker [12].

In our present study, highly purified human renal proximal tubular epithelial cells were used as an in vitro model to study the cellular response to an inflammatory microenvironment. Therefore, a cytokine-induced inflammatory system was established to analyze different miRNA expression levels in cells and their EVs. Specifically, we describe and discuss the altered miRNA expression of PTC and their released EVs during induced inflammation. Especially with regard to a possible clinical use of miRs as new biomarkers, the accurate characterization of the miR expression altered during inflammatory processes is of enormous importance.

## 2. Results

### 2.1. Characterization of PTCs and Their EVs

Cultured PTCs displayed an epithelial morphology with a highly compact cell monolayer (Figure 1A). The expression of the characteristic markers CD13 and CD26 was shown via flow cytometry. In addition, the expression of the epithelial cell adhesion molecule (EpCAM) and the marker CD63, a characteristic marker of exosomes, was shown (Figure 1B). After incubation in an inflammatory microenvironment, the cells retained their epithelial morphology but showed smaller gaps at some sites of the cell monolayer.

SEC-isolated PTC-EVs were characterized via nanoparticle tracking analysis (NTA) (Figure 1C,E) and flow cytometry (Figure 1D). EVs isolated from unstimulated controls showed a mean size of 173.7 ± 24.2 nm (distribution between 24 and 484 nm), with an average concentration of 8.47 × 10^10^ particles/mL (*n* = 3, Figure 1C). EVs isolated from stimulated cells were slightly larger in their average mean size (192.3 ± 23.8 nm, distribution between 60 and 576 nm), and with a slightly increased concentration (9.48 × 10^10^ particles/mL, *n* = 3, Figure 1E). Nevertheless, these differences were not significant. Flow cytometric analyses further showed CD63 expression on magnetic-bead-purified PTC-EVs (using specific EpCAM-marked magnetic beads) (Figure 1D).

Additional qPCR analyses to validate the effect of an inflammatory microenvironment on the cells showed significantly upregulated mRNA expression levels of interleukin (IL) 6 (IL6), IL1β, and intercellular adhesion molecule 1 (ICAM1) (Figure 2A–C). For these experiments, we used a total of six PTC samples: two identical purifications that were also used for miR sequencing and four additional cell cultures.

### 2.2. Effect of Inflammation on the miR Expression in PTC and Their EVs

To sequence all samples, the high-throughput Illumina sequencing technology together with the Miseq instrument were used, resulting in the generation of raw DNA sequence reads. In this study, a total of around 9,000,000 raw reads were obtained from miRNAs and the slightly longer so-called piwi-interacting RNAs **(**piRNAs) of PTCs or of EVs released from PTCs, respectively. Nevertheless, the actual number of reads was approximately 1,500,000 reads for samples obtained from the PTCs and 50,000 reads for samples obtained from the EVs.

To finally determine the identity and quantity of all reads included in each sample, the raw data of the sequencing were uploaded to the Qiagen database for further analysis. Therefore, the specific miRNAs and piRNAs that corresponded to the raw reads generated from Illumina sequencing were evaluated. Hierarchical clustering in the form of a heatmap should give an overview about the differences in the quantities of the different miRNAs and piRNAs between the unstimulated controls and the stimulated samples. Those miRNAs and piRNAs that had a minimum log fold change (log_2_FC) of >±0.75 and an adjusted *p* value (p-adj) of less than 0.1 were each shown in a heatmap as up- or downregulated (Figure 3A,B, Appendix A). Of the detected small RNAs from PTCs, there were 13 upregulated and 10 downregulated (Figure 3A, Appendix A). Of the detected miRNAs and piRNAs from the EV samples, there were 16 upregulated and 34 downregulated (Figure 3B, Appendix A).

For further analyses, we only used highly significant regulated miRNAs (p-adj < 0.05) with a minimum log_2_FC of >±1, and no piRNAs (Figure 4A,B and Figure 5). We found 12 miRNAs in the PTC samples, of which 6 (miR-146A-5p, miR-147b, miR-146a-3p, miR-155-5p, miR-99b-5p, and miR-100-3p) were upregulated and 6 were (miR-210-3p, miR-128-3p, miR-186-5p, miR-335-5p, miR-140-3p, and miR-296-3) downregulated (Figure 4A), and 9 miRNAs in the EVs from PTCs, of which 8 were upregulated (miR-146a-5p, miR-155-5p, miR-141-3p, miR-221-3p, miR-23b-3p, miR-147b, miR-320c, and miR-3613-5p) and 1 was downregulated (miR-3687) (Figure 4B).

The Venn diagram shows the overlap of the highly significant miRNAs found in PTCs (12 miRNAs) and their EVs (9 miRNAs) (Figure 5A). These three miRNAs (miR-146a-5p (log_2_FC 3.384 (PTC) and 1.944 (EVs)), miR-147b (log_2_FC 4.244 (PTC) and 3.007 (EVs)), and miR-155-5p (log_2_FC 2.438 (PTC) and 1.232 (EVs)), also shown in Appendix A) were then used for network analyses to identify the contribution of these miRNAs to biological processes and target interactions (using miRNet 2.0, miRTarBase v8.0) (Figure 5B,C). In total, the three miRNAs were predicted to be involved in 55 pathways, including the Toll-like receptor signaling pathway (p-adj < 0.0000305), the T cell receptor signaling pathway (p-adj < 0.0000404), apoptosis (p-adj < 0.00142), and the B cell receptor signaling pathway (p-adj < 0.00677), and were predicted to target 3775 genes (Figure 5B,C and Appendix A). Shared targets in the Toll-like receptor signaling pathway for miR-146a-5p and miR-155-5p were IL6, ICAM1, NF*k*B1, FADD, RHOA, RAC1, CXCL8, STAT1, and FOS and COL4A2 for miR-147b and miR-155-5, respectively.

The expression of the three miRNAs found (Figure 5A) was then further validated using the purchasable miRCURY system from Qiagen. Using this system, we confirmed the highly significant upregulation of all three miRNAs after the incubation of PTCs in the inflammatory microenvironment (Figure 6). Nevertheless, due to a lack of samples, we did not validate the expression of the miRNAs in EVs.

Finally, we examined whether the three miRNAs found correlated with the levels of the three inflammatory molecules from the PCR of the associated samples (see Figure 2). This showed that both IL6 and ICAM correlated well with the reads of the three miRNAs (for all correlations: r > 0.89, *p* < 0.05) (Figure 7A,B). Nevertheless, this was not the case for IL1β (*p* not significant for all the correlations).

## 3. Discussion

Understanding the precise molecular and cellular mechanisms that lead to inflammation in the kidney is the most important way to identify early targets for the prevention or treatment of acute kidney injury. Inflammation is a physiological process to protect the body from acute injury induced, for example, by toxins or pathogens. It is mediated by inflammatory cytokines, such as IL1β, IFNγ, and TNFα, released by immune cells, but also by resident tissue cells like PTCs. Thus, PTCs are involved in physiological and pathophysiological processes. The transmission of intercellular signal molecules is of great significance, including endocrine, paracrine, and autocrine biochemical pathways. Following injury or the stimulation of tubular epithelial cells, various cascades of mediator systems may be activated, leading to an increased local release of cytokines, chemokines, and other pro-inflammatory molecules. All these locally produced mediators may subsequently lead to injury enhancement, either directly or indirectly through increased the immigration of proinflammatory cells such as macrophages and lymphocytes [13]. In addition, the expression of various intracellular non-coding RNAs (miRNAs, long non-coding RNAs, and piRNAs) is altered, and extracellular vesicles as mediators of cell–cell communication are released with a cargo of inflammation-specific molecules [5,14]. Although the mechanisms regulating inflammation and the subsequent defense mechanism have not been fully elucidated, recent evidence suggests that non-coding RNAs, particularly miRNAs, play a critical role in the generation and control of the inflammatory response. These non-coding RNAs are critical for the stability and maintenance of mRNA at a post-transcriptional level. Several miRNAs, such as miR-146 and miR-155, have been shown to be important regulators of inflammation-related mediators [15]. Alexander and co-workers showed that miR-146a and miR-155 are present in exosomes and pass between immune cells in vivo and demonstrated that exosomal miR-146a inhibits while miR-155 promotes endotoxin-induced inflammation in mice [15].

There are currently limited data on altered miRNA expression in renal epithelial cells and their EVs during episodes of renal inflammation in general. Nevertheless, several studies investigated the involvement of miRNA during injury and repair in the kidney [16]. Recent studies have shown that miRNAs play a critical role in the development of acute renal failure. The overexpression of miR-150 in mice generated renal cell injury due to the downregulation of a growth factor receptor and the consequent enhancement of inflammation and the apoptosis of interstitial cells [17]. The upregulation of miR-687 in renal tubular epithelial cells both in vitro and in vivo induced renal injury by suppressing the expression of specific molecules, the activation of the cell cycle, and the induction of apoptosis [18]. On the other hand, the upregulation of miR-21 has been shown to ameliorate induced kidney injury by inhibiting inflammation and cell apoptosis [19]. Joo and co-workers demonstrated that miR-125b expression was enhanced in injury and protected the kidney from cisplatin-induced kidney injury [20]. Another study suggested that miR-155 had a protective role in cisplatin-induced nephrotoxicity [21]. The induction of miR-26a has been shown to increase tubular cell viability through the modulation of a growth and a transcription factor during kidney injury in an in vivo model [22]. Several other studies showed the influence of specific miR on cell viability and proliferation during renal regeneration (reviewed in [16]).

EVs released by (renal) cells are important mediators of cell–cell-communication and signal transduction. Urinary EVs have been intensely investigated and characterized for use as diagnostic non-invasive biomarkers of inflammatory renal diseases [23]. However, their specific cellular origin is unclear, as they could originate from any of several cell types from the kidney and bladder. For this reason, the accurate characterization of altered miRNA expression in inflammatory processes in a well-defined system of highly purified and differentiated PTCs and their released EVs is necessary. For example, Wang and co-workers identified differential expression levels of miRs and proteins from PTC-derived exosomes, a specific subpopulation of EVs, under different disease-culture conditions and described different readouts linked to renal pathological processes [14]. They identified different miRs differently expressed in exosomes between the control and the inflammatory group, such as miR-200a, miR-222, and miR-204, which were previously attributed a role within the renal disease pathways [14]. Another study by the same authors used a Transwell system to examine the molecular content and function of EVs released from the apical and the basolateral surface of polarized human primary tubular epithelial cells under inflammatory diseased conditions [5]. They used multiomic analysis to characterize the distinct molecular profiles of miRs, proteins, and lipids released from EVs of the apical versus basolateral cell membranes. The study demonstrated that EVs released, particularly basolaterally, play a central role in modulating tubulointerstitial inflammatory responses observed in immune-mediated renal disease.

Our data revealed that miR 146a, miR-155, and miR-147b were significantly upregulated in both PTCs and EVs after inflammatory stimulation. All three miRNAs have been implicated in inflammatory processes and are directly involved in the Toll-like receptor signaling pathway [24,25]. Following this, the expression of the three microRNAs has been found increased in patients with bacterial sepsis [26]. Whilst it has been proposed that the overexpression of miR-147b inhibited LPS-induced inflammation via the inhibition of the p38 MAPK signaling pathway and alleviates inflammation and apoptosis in acute lung injury, miR-146a and miR-155 have emerged as important transcriptional regulators in the regulation of NF-κB-driven inflammation [27,28]. LPS-induced inflammation in human microvascular endothelial cells showed increased levels of IL6 and ICAM1 together with an upregulation of miR-146a [29]. The overexpression of miR-146a resulted in the downregulation of IL6, as well as ICAM1, while the inhibition of miR-146a led to opposite effects. In this regard, Zhou and co-workers reported a positive feedback loop between miR-155-5p, STAT3, and NF-κB in A. cantonensis infection and the miR-155-5p-dependent expression of eosinophil inflammatory cytokines such as CCL6/C10, ICAM1, and MMP9 [30]. In inflammatory responses, the NF-κB-miR-155 axis and the NF-κB-miR-146a axis regulate the intensity and duration of inflammation [31]. As one of the first miRNAs identified as an immune system regulator, miR-146 is rapidly expressed upon NF-κB activation and downregulates IRAK1 and TRAF6 to reduce the activity of NF-κB by a negative regulatory loop mechanism [32]. miR-146a is involved in the pathogenesis and progression of several immune inflammatory diseases such as rheumatoid arthritis, inflammatory bowel disease, multiple sclerosis, psoriasis, atherosclerosis, hepatitis, or chronic obstructive pulmonary disease.

Like miR-146, miR-155 also plays a critical role in inflammatory diseases such as inflammatory autoimmune diseases (e.g., rheumatoid arthritis, systemic lupus erythematosus, multiple sclerosis, type 1 diabetes, and systemic sclerosis) and is abnormally expressed in inflammatory bowel disease, colitis ulcerosa, and neuroinflammation [33]. miR-155 is rapidly upregulated by NF-κB within the inflammatory response. Once activated, miR-155 controls the expression of both IKKβ and IKKε, forming a positive feedback loop, which leads to the repression of NF-κB activation [31]. Thus, miR-155 and miR-146 regulate the inflammatory response in a combinatorial manner [34]. Clinical and preclinical studies have shown that renal miR-155 expression increases significantly in acute renal failure [35]. An in vitro study has shown that the overexpression of miR-155 promoted cellular apoptosis and suppressed proliferation, while the inhibition of miR-155 expression exerted opposite effects [36]. For this reason, we additionally examined the correlation of the miRs found with the molecules IL6, IL1β, and ICAM1, which are increased by inflammation, and we were able to show that all three miRs correlate with the expression of IL6 and ICAM1, but not with that of IL1β.

miR-146a is also involved in the regulation of inflammation in the kidney. Fu and co-workers reported that miR-146a reduced lupus-erythematosus-induced renal injury in MRL/lpr mice by downregulating the NF-κB pathway [37]. In addition, a protective effect of miR-146 against kidney injury in diabetic nephropathy rats through mediating the NF-κB signaling pathway has also been shown [38]. It is tempting to speculate about the role of miR-147b in renal inflammation. Interestingly, a recent publication described renal injury due to the induction of miR-147 during cold-storage-associated transplantation [39]. miR-147 suppressed NDUFA4, resulting in mitochondrial damage and renal tubule cell death.

In summary, our current study provides further insight into the inflammatory response of renal proximal tubular epithelial cells. We characterized their altered expression of small RNAs in response to cytokine stimulation and compared it with their altered cargo to released EVs. We found three characteristic miRNAs, miR-146a, miR-147b, and miR-155, which were significantly upregulated in cells and in EVs, and thus may play a critical role in regulating the inflammatory response in the kidney.

## 4. Materials and Methods

### 4.1. Isolation and Culture of Human Renal Proximal Tubular Epithelial Cells

Human renal proximal tubular epithelial cells (PTCs) were separated as described previously [40]. Briefly, PTCs were isolated after nephrectomies from tissue not involved in renal cell carcinoma. Kidney tissue was minced with crossed blades, digested with collagenase/dispase, and forced through a sieve (106 µm). The tissue slurry was then incubated with collagenase IV, DNase, and MgCl2 and centrifuged over a Percoll density gradient. PTCs were then isolated in high purity using a mAb against aminopeptidase M (CD13) and the Mini-MACS system (Miltenyi, Bergisch Gladbach, Germany). Cells were characterized as previously described [40,41], seeded in cell culture plates, and cultured in standard culture medium (Medium 199 (M4530, Sigma, Taufkirchen, Germany) with a physiologic glucose concentration (100 mg/dL) and with 10% fetal bovine serum (FBS; Biochrom, Berlin, Germany)). The medium was replaced every three to four days, and confluent cells were passaged via trypsinization. Cells isolated from four different patients were used. Cells between passages 3 and 6 were used for the experiments. Selected PTCs were used for further characterizations via flow cytometry, as previously described [40].

### 4.2. Preconditioning in an Inflammatory Microenvironment

Cells were grown in culture flasks (75cm^2^) in standard cell culture medium until confluence. The cells were then washed and kept in serum-free medium 199 for 48 h, either in standard medium (controls) or in an inflammatory microenvironment. The inflammatory microenvironment was induced by culturing PTCs in medium with a mixture of cytokines (CM) containing γ-interferon (200 U/mL), interleukin-1β (25 U/mL), and TNF-α (10 ng/mL). Then, PTCs were used for direct miR isolation (see Section 4.5), and EVs were isolated from the supernatant via size exclusion chromatography (SEC) (see Section 4.3).

In addition, the effect of the inflammatory microenvironment was checked via qPCR analysis. The complete method was described earlier [42]. The primers used were IL6 (150 bp, forward AAA GAT GGC TGA AAA AGA TGG ATG C, reverse ACA GCT CTG GCT TGT TCC TCA CTA C), IL1β (83 bp, forward AGC TGA TGG CCC TAA ACA GA, reverse AGA TTC GTA GCT GGA TGC CG), ICAM1 (135 bp, forward CAA CCT CAG CCT CGC TAT GG, reverse CGG GGC AGG ATG ACT TTT GA).

### 4.3. Isolation of Extracellular Vesicles

The preconditioned supernatant was used to isolate EVs from cells cultured under normal and inflammatory pretreatment for 48 h (growth area of 150 cm^2^ with 16 mL of serum-free DMEM). After 48 h of incubation, the medium was centrifuged for 10 min at 600× *g* in order to remove cell debris, and then filtered using a 0.45 µm PVDF filter and concentrated approximately 40-fold via centrifugation at 2800× *g* for 20 min using a Vivaspin^®^ Turbo 30 MWCO Centrifugal Filter. The concentrated medium was then used for the isolation of EVs via SEC using Sepharose CL-2B columns [43,44]. The concentrated medium was applied onto a Sepharose CL-2B column, which was washed and equilibrated with PBS. As elution buffer, PBS was applied to the column until 18 flow-through fractions with a respective volume of 500 µL were collected. The fractions 7–12 containing extracellular vesicles were pooled and subsequently concentrated (to approximately 400 µL EV solution) using a 3 kDa molecular weight cut-off Amicon filter by centrifuging for 25 min at 2800× *g*. EV samples were used immediately for characterization or miR isolation. Selected EV isolations were used for further characterizations.

### 4.4. Characterization of Extracellular Vesicles

We used nanoparticle tracking analysis (NTA) to determine the size distribution and concentration of the isolated EVs, as previously described [42]. In brief, 10 µL of concentrated EV solution was diluted 1:100 and immediately analyzed using a NanoSight NS500 (Malvern Panalytical, Malvern, UK) according to the manufacturer’s instructions (camera settings: level 14, camera gain 1.5, and temperature 28 °C). Videos were analyzed with the Nano Sight NTA 3.2 software (threshold 14, gain 1.5).

To further analyze PTC-EVs via flow cytometry, a commercially available Flow Detection Magnetic Bead Reagent Kit was used (Invitrogen, Carlsbad, CA, USA, No. 10618D (anti-EpCAM)). In brief, a 40 µL anti-EpCAM magnetic bead solution was added to 1 mL of isolation buffer (PBS, 2% FCS, 1 mM EDTA, 0.1% Sodium azide) and placed on a magnetic stand for 2 min. Then, the supernatant was removed, and the bound beads were washed and re-suspended in 90 µL of isolation buffer. Then, a 20 µL SEC-isolated PTC-EV solution in PBS was added and incubated overnight at 4 °C. The sample was then washed twice with 1 mL of isolation buffer before the beads were re-suspended in isolation buffer (300 µL). The bead-EV solution (100 µL) was then stained with 20 µL of APC-labeled anti-CD63 detection antibody (ImmunoTools, Friesoythe, Germany) and incubated under rotation at RT in the dark for 45 min and washed and re-suspended in a 300 µL isolation buffer. The samples were measured using a flow cytometer (BD Biosciences, Heidelberg, Germany) and analyzed on the instrument until 10,000 events were detected. As a control for the fluorescent measurements, a 100 µL bead-EV solution without detection antibody was mixed with 200 µL of isolation buffer and measured. In addition, we used commercially available Dynabeads M-280 (Fisher Scientific, Schwerte, Germany, No. 11205D) to calibrate forward and sideward scatter in the flow cytometric measurement.

### 4.5. Isolation of miR from PTC

For miR extraction from cultured PTC, the NucleoSpin^®^ miRNA kit (Macherey-Nagel, Düren, Germany) was used in accordance with the manufactures protocol. First, cells were harvested by adding 1.0 mL of Nucleozol to each flask (75 cm^2^) and then scratching all cells from the bottom. The mixture was then transferred into a new 1.5 mL tube, and 400 μL of RNase free water was added to each sample and mixed by shaking the tube vigorously multiple times. Afterwards, the mixture was incubated for 5 min at RT. The samples were centrifugated at 12,000× *g* for 15 min at RT. Then, 300 μL of each supernatant was transferred to a new tube, and 100 μL of 100% ethanol was added and incubated for 5 min at RT. Afterwards, a NucleoSpin RNA Column was placed into a 2 mL tube and the supernatants were loaded onto the column. The supernatant was collected for further miRNA extraction. Next, 350 μL of buffer was added to the column and the tubes were centrifugated for 1 min at 11,000× *g*. The flow-through was discarded, and 100 μL of rDNase was added to the column to digest DNA. The samples were incubated for around 15 min at RT. In the meantime, the flow-through containing small RNAs was used for adjusting binding conditions for small RNAs by adding 300 μL of Buffer MP to each flow-through. After vortexing the samples for 5 s, the tubes were centrifuged for 3 min at 11,000× *g* to pellet the protein. To remove the protein precipitate, the samples were loaded onto a NucleoSpin Protein Removal Column in a 2 mL tube. After a centrifugation step for 1 min at 11,000× *g*, the column was discarded, 800 μL of buffer was added to each flow-through, and the samples were vortexed for 5 s. To bind small RNA, for each sample, a new NucleoSpin RNA Column was placed in a 2 mL tube, and 725 μL of the sample was loaded onto the column. The tubes were centrifuged for 30 s at 11,000× *g*, and the flow-through was discarded. This step was repeated until the whole sample was loaded onto the column. In the next step, the columns were washed by adding 600 μL of buffer. After a centrifugation step for 30 s at 11,000× *g*, the flow-through was discarded, and afterwards, 700 μL of buffer was added to each sample. Again, the tube was centrifugated at 11,000× *g* for 30 s and, 250 μL of buffer was added and the tube was centrifugated for 2 min at 11,000× *g* to remove all residuals from the membrane. The column was transferred to a new collection tube, and 15–30 μL of nuclease-free water was added on the center of the membrane. After incubation for 1 min at RT, a centrifugation step followed for 1 min at 11,000× *g* to recover the miR.

### 4.6. Isolation of miR from EVs

Whole miR was isolated from EVs using the NuceloSpin^®^ miRNA Plasma Kit (Macherey-Nagel, Düren, Germany) according to the manufacturer’s instructions. Therefore, 90 μL of buffer was added to 300 μL of the sample and mixed by vortexing the tube for 5 s. The mixture was incubated for 3 min at RT, and in connection, 30 μL of buffer was added. The sample was mixed by vortexing it for 5 s, and it was incubated for 1 min at RT. Afterwards, the tube was centrifugated at 11,000× *g* for 3 min at RT. The clear supernatant was transferred to a new 2 mL tube and mixed with 400 μL of isopropanol. The mixture was then transferred onto an miRNA column which was placed into a collection tube and centrifugated at 11,000× *g* for 2 min at RT. The flow-through was discarded, and 100 μL of buffer was applied on the column. Again, the tube was centrifugated at 11,000× *g* for 30 s at RT, and 700 μL of buffer was added. After another centrifugation step at 11,000× *g* for 30 s at RT, the flow-through was discarded, and the washing step was repeated with 250 μL of buffer, and the tube was centrifugated for 2 min at 11,000× *g* to remove all residuals from the membrane. The column was transferred to a new collection tube, and 30 μL of nuclease-free water was added on the center of the membrane. After a centrifugation step for 1 min at 11,000× *g* to recover the miR, the tube was incubated for 1 min at RT.

### 4.7. miR Sequencing

The concentration of miR was analyzed using the HS RNA Kit for TapeStation 4150 (Agilent, Waldbronn, Germany). The miR library was prepared using the QIAseq miRNA Library Kit according to the manufacturer’s instructions. After library preparation, the DNA concentration was analyzed by using a Qubit dsDNA Assay Kit in Qubit 3.0 Fluorometer (ThermoFisher Scientific, Darmstadt, Germany). In addition, the DNA quality was checked using an HS DNA Kit of the Bioanalyzer 2100 (Agilent, Waldbronn, Germany). Next-generation sequencing (NGS) was performed using the MiSeq Reagent Kit v3 (Illumina, San Diego, CA, USA), PhiX Sequencing Control v3, and MiSeqTM Desktop Sequencer. For this, samples were pre-diluted (1:5 or 1:10) and prepared for sequencing according to the manufacturer’s instructions. Samples were loaded into the MiSeqTM Desktop Sequencer in a sequencing cassette, and sequencing was performed.

Coverage files were converted into raw count matrices using the Qiagen pipeline. A priori filtering for sparse read counts was applied. The bioinformatics tool iDEP.96 was used to statistically analyze and display the miR sequencing data [45]. We performed a hierarchical cluster analysis of raw count data using the normalized read counts of the DESeq2 package. Kyoto encyclopedia of genes and genomes (KEGG) pathway analysis based on microRNA signature was performed with miRNet (miRTarBase v8.0) [46] for all the identified miRNAs. The Venn diagram was created in Venny 2.1.0 [47].

### 4.8. Validation of miR Expression

For the validation of selected miRNAs, cDNA synthesis was performed using the miRCURY LNA miR Assay Kit (339340, Qiagen, Venlo, The Netherlands) according to the manufacturer’s protocol. In brief, a master mix was prepared on ice containing 5 μL 2× miRCURY SYBR^®^ Green Master Mix, 0.5 μL ROX Reference Dye, 1 μL PCR primer (miR-146a-5p, No. YP00204688; miR-147b, No. YP00204368; miR-155-5p, No. YP02104687; all from Qiagen) and 1 μL of RNase free water per sample. After vortexing, 7 μL of the Master mix and 3 μL of the cDNA template were loaded onto a 96-well PCR plate. For the no-template controls, 3 μL of RNase free water was used. Quantitative PCR was carried out under the following conditions: two minutes at 95 °C for enzyme activation, then 10 s at 95 °C and 60 s at 56 °C for 40 cycles. For the quantification of the PCR fragments, we used the ABI Prism^®^ 7900HT Fast Real-Time PCR System with a Sequence Detection System SDS 2.4.1 (Thermo Fisher Scientific, Darmstadt, Germany). Relative quantification was carried out with the ∆∆CT method using spike-in control UniSP6 (Qiagen, No. YP00203954) for normalization [48], and the level of target gene expression was calculated using 2^−ΔΔCt^.

### 4.9. Statistical Analysis

For the analysis of qPCR data, we used Student’s *t*-test. The data are expressed as mean ± SEM. *p* values < 0.05 were considered significant. The statistical analysis and Pearson r value correlation of the measured data as well as their graphic representation was performed with the software GraphPad Prism 7.04.

## Figures and Tables

**Figure 1 ijms-24-11069-f001:**
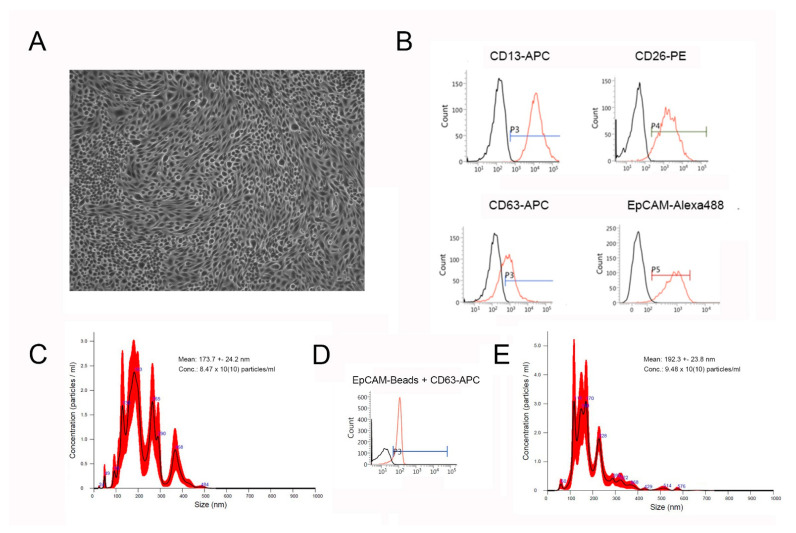
Characterization of PTCs (**A**,**B**) and EVs isolated from PTC supernatant (**C**,**D**). (**A**) Characteristic phase contrast microscopy of confluent PTCs cultured in standard cell culture. (**B**) Representative flow cytometric overlay histograms of characteristic PTC marker expression (CD13, CD26, and EpCAM), and of CD63, a characteristic marker of exosomes. The black histograms represent isotype controls. (**C**) Representative nanoparticle tracking analysis (NTA) of EVs isolated from unstimulated PTC after standard cell culture for 48 h. (**D**) Representative flow cytometric overlay histogram of PTC-EVs isolated using EpCAM-beads and immunostained with CD63-APC. The black histogram represents an unstained control. (**E**) Representative nanoparticle tracking analysis (NTA) of EVs isolated from PTCs after culture in an inflammatory microenvironment for 48 h.

**Figure 2 ijms-24-11069-f002:**
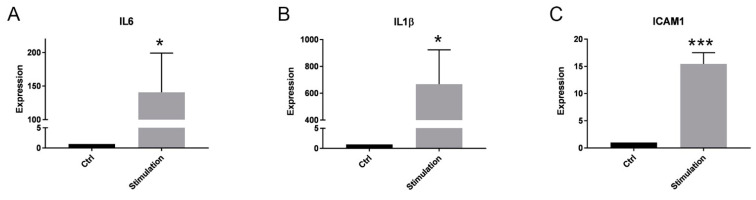
Validation of inflammatory stimulation of PTCs. mRNA expression of selected factors under an inflammatory culture condition for 48 h: (**A**) IL6, (**B**) IL1β, and (**C**) ICAM1. The expression levels in each experiment were normalized to a housekeeping gene (β-actin) and expressed relative to the control using the ∆∆CT method (Expression). A mixture of pro-inflammatory cytokines (stimulation) significantly increased the mRNA expression of all three readouts after 48 h of incubation. Mean ± SEM, * *p* < 0.05 (A, B), *** *p* < 0.001 (C), *n* = 6.

**Figure 3 ijms-24-11069-f003:**
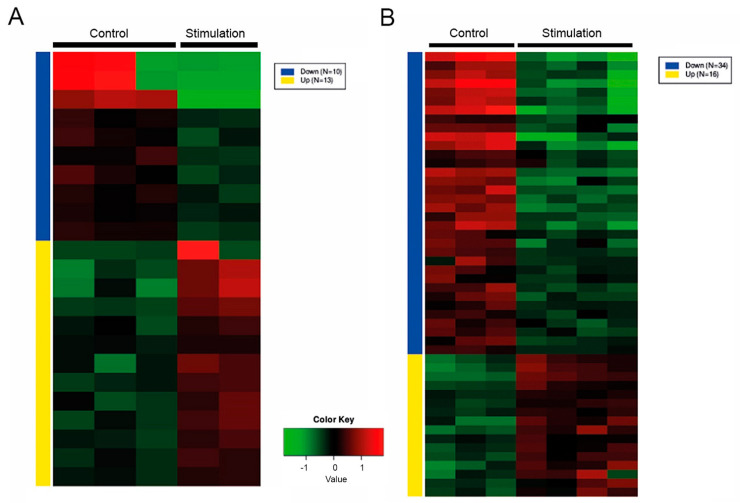
RNAseq analysis of miRNA extracted from PTCs (**A**) and EVs (**B**). Heatmap showing hierarchical clustering of analyzed miRNAs and piRNA from stimulated PTCs or their EVs and unstimulated controls (minimum fold change ±1.5, p-adj < 0.1). The groups are represented in columns and the specific miRNAs in rows. Each colored cell on the map corresponds to an expression value.

**Figure 4 ijms-24-11069-f004:**
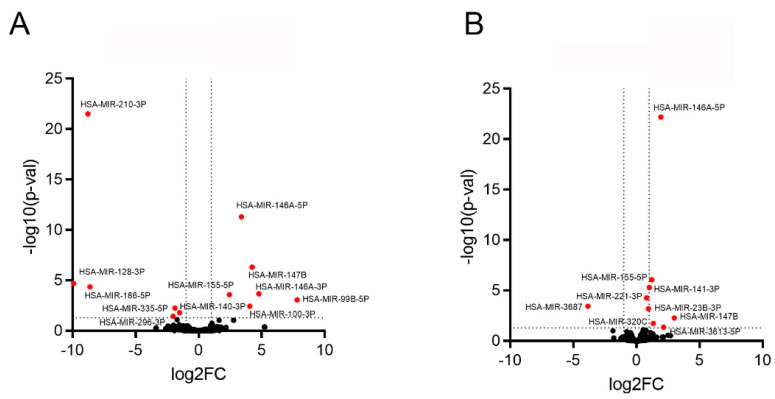
RNAseq analysis of miRNA extracted from (**A**) PTCs and (**B**) their EVs. Volcano Plot combining log fold change (log_2_FC) analysis. Fold changes with a highly significant upregulation or downregulation are highlighted (log_2_FC > ±1; p-adj < 0.05).

**Figure 5 ijms-24-11069-f005:**
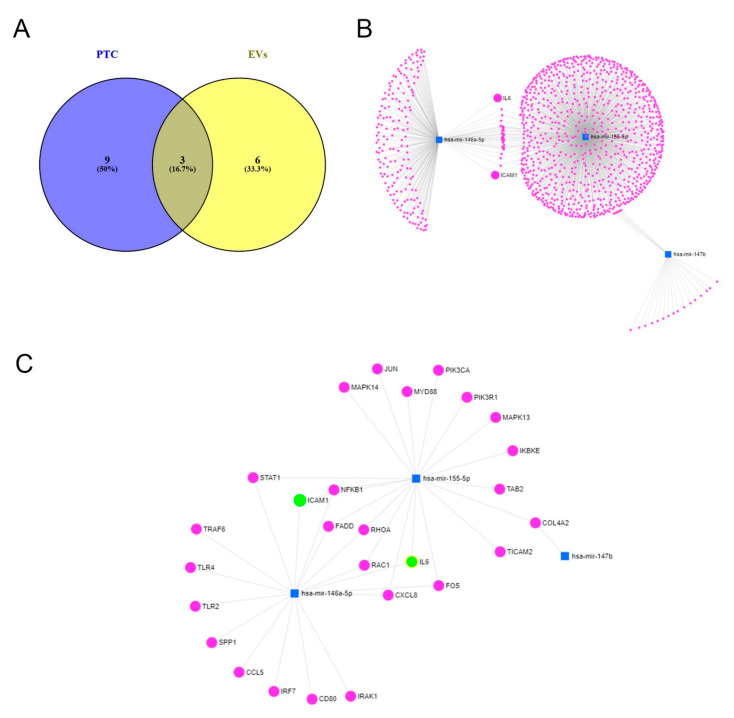
Venn diagram and network analysis. (**A**) Venn diagram to show the overlapping highly significant regulated miRNA from PTCs and their EVs. Only three miRNAs (miR-146a-5p, miR-147b, and miR-155-5p) overlapped between the two groups (Venny 2.1.0). (**B**,**C**) Network analyses showing miRNA target interactions. Network visualization: miR (blue squares) and the potentially interacting gene network (magenta and green dots) created with miRNet 2.0 (miRTarBase). (**B**) Dissected miRNA–target interactions and functional associations through network-based visual analysis (miRNet). miR-146a-5p, miR-155-5p, and miR-147b regulate 1092 genes, including ICAM1 and IL6. (**C**) Interaction and targets involved in Toll-like receptor signaling pathway.

**Figure 6 ijms-24-11069-f006:**
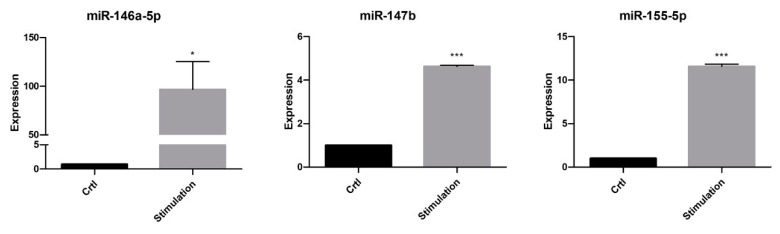
Validation of miR-146a-5p, miR-147b, and miR-155-5p expression in unstimulated PTCs (controls) and after stimulation. Values were normalized using the UniSP6 spike-in control primer set. Data are mean  ±  SEM of three independent experiments. Significance was calculated using Student’s *t*-test. * *p*  <  0.05; *** *p*  <  0.001.

**Figure 7 ijms-24-11069-f007:**
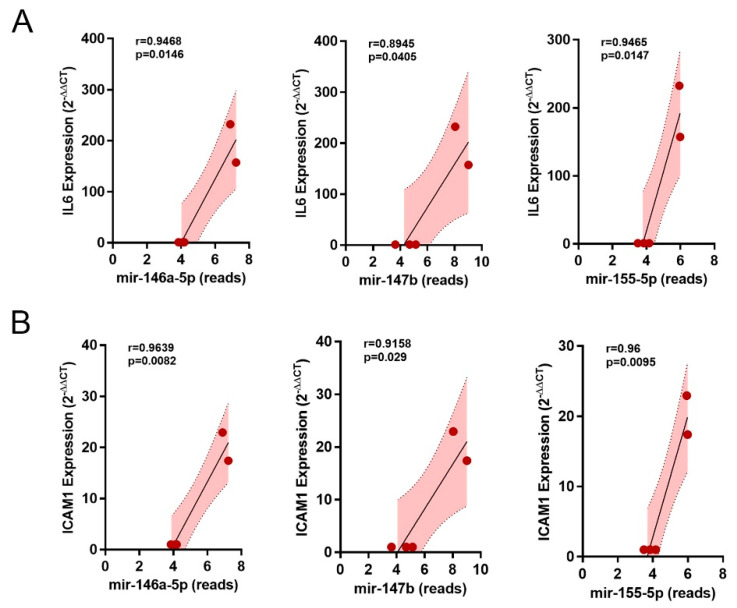
Correlations. Correlation of overlapping miRNAs (reads of miR-146a-5p, miR-147b, and miR-155-5p) with (**A**) IL6- and (**B**) ICAM1-mRNA expression.

## Data Availability

Not applicable.

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
