# Peer review of "microRNA Expression of Renal Proximal Tubular Epithelial Cells and Their Extracellular Vesicles in an Inflammatory Microenvironment In Vitro"

_ijms, 2023, doi:10.3390/ijms241311069_

Round 1

Reviewer 1 Report

The manuscript „Microrna Expression of Renal Proximal Tubular Epithelial Cells 2 and Their Extracellular Vesicles in an Inflammatory Microenvironment in Vitro” by Baer et al. is an excellent, well written work investigating the changes of miRNA expression in human Renal Proximal Tubular Epithelial cells (PTC) and in their released extracellular vesicles (EV) under inflammatory conditions.

The manuscript is in my opinion of great interest for readers of IJMS.

I have only few observations:

1) the cells are cultivated for 48 h in serum-free medium before miRNA isolation. Is there any reason for this? Serum-free incubation should stimulate autophagocytotic processes and apoptosis, which may be confounding factors in the analysis.

2) according to the description of the Materials and Methods, EV have been isolated from the supernatant of cultivated PTC cells, corresponding to the luminal membrane of proximal tubules. Therefore, these EVs are probably a model for EVs released into the urine. Are the cells from the renal proximal tubules able to release EVs from the basolateral membrane domain into the blood? Can be such a basolateral EVs release much more important under inflammatory conditions as a cross-talk with other organs?

Minor:

3) Line 136: please explain what is piRNA

4) Figure 3: please explain  the meaning of red and green colors

5) line 155: …and 6 were….

6) line 297: please explain what is SLE

7) line 391: …supernatant….

8) line  416: …The sample was ?.....

9) line 447: …The Venn diagram….

Author Response

REVIEWER 1

The manuscript „MicroRNA Expression of Renal Proximal Tubular Epithelial Cells 2 and Their Extracellular Vesicles in an Inflammatory Microenvironment in Vitro” by Baer et al. is an excellent, well written work investigating the changes of miRNA expression in human Renal Proximal Tubular Epithelial cells (PTC) and in their released extracellular vesicles (EV) under inflammatory conditions. The manuscript is in my opinion of great interest for readers of IJMS.

We thank the reviewer for the valuable comments to improve the manuscript and have now edited and replied all notes and criticisms (as a point-by-point reply).

I have only few observations:

1) the cells are cultivated for 48 h in serum-free medium before miRNA isolation. Is there any reason for this? Serum-free incubation should stimulate autophagocytotic processes and apoptosis, which may be confounding factors in the analysis.

Answer: The cells were incubated in serum-free conditions, because serum already contains many EVs of blood cells. The complete removal of these EVs from serum is very difficult and is often not complete. Therefore, the results of the miR-Seq would thus be distorted.

2) according to the description of the Materials and Methods, EV have been isolated from the supernatant of cultivated PTC cells, corresponding to the luminal membrane of proximal tubules. Therefore, these EVs are probably a model for EVs released into the urine. Are the cells from the renal proximal tubules able to release EVs from the basolateral membrane domain into the blood? Can be such a basolateral EVs release much more important under inflammatory conditions as a cross-talk with other organs?

Answer: This is absolutely correct. By standard culture on normal cell culture plastic, the EVs released are like urinary EVs. It is also possible to culture the cells on membrane inserts, where the two-chamber culture system mimics the nephronal epithelia with a luminal and a basal side. The problem with this culture system is the unavailable size of cell culture inserts to provide enough medium for EV isolation and thus miR isolation. We therefore decided to use the standard cell culture system on plastics. In principle, a more detailed investigation of the release across the luminal and basal cell membrane would of course be desirable

Minor:

3) Line 136: please explain what is piRNA

We have now explained the abbreviation piRNA (in line 139).

4) Figure 3: please explain the meaning of red and green colors

We have added a color key to the figure.

5) line 155: …and 6 were….

This has now been corrected.

6) line 297: please explain what is SLE

We have now changed the abbreviation “SLE” to lupus erythematosus.

7) line 391: …supernatant….

This has now been corrected.

8) line  416: …The sample was ?.....

This has now been corrected.

9) line 447: …The Venn diagram….

This has now been corrected.

Reviewer 2 Report

Renal proximal tubular epithelial cells (PTC) are central in kidney inflammation. They respond to inflammatory signals, altering their mRNA, microRNA, protein, and lipid expression, and release inflammation-specific extracellular vesicles (EVs). Understanding these mechanisms is crucial for treating acute kidney injury. Using purified human PTCs as a model, miRNA expression in cells and EVs was analyzed. Results show significant regulation of miRNAs in PTC and EVs, with only three overlapping, which are involved in regulating inflammation pathways. The work was done at a high methodological level, but a number of changes and additions are required for its further publication.

This work was performed on human PTC cells, however, it is necessary to indicate how many individual cultures from patients were included, the age of the patients, and their main diagnosis.

It is essential to describe the physiological state of the cells after modeling the inflammatory effect, compare the amounts of production of EVs and their size distribution between cultures before and after the modeling of inflammation.

The research article identified various miRNAs through sequencing. However, to enhance the understanding of the molecular mechanisms of inflammation in PTC, a differential analysis of miRNA expression using the PCR method should be performed.

Author Response

REVIEWER 2

Renal proximal tubular epithelial cells (PTC) are central in kidney inflammation. They respond to inflammatory signals, altering their mRNA, microRNA, protein, and lipid expression, and release inflammation-specific extracellular vesicles (EVs). Understanding these mechanisms is crucial for treating acute kidney injury. Using purified human PTCs as a model, miRNA expression in cells and EVs was analyzed. Results show significant regulation of miRNAs in PTC and EVs, with only three overlapping, which are involved in regulating inflammation pathways. The work was done at a high methodological level, but a number of changes and additions are required for its further publication.

We thank the reviewer for the valuable comments to improve the manuscript and have now edited and replied all notes and criticisms (as a point-by-point reply).

This work was performed on human PTC cells, however, it is necessary to indicate how many individual cultures from patients were included, the age of the patients, and their main diagnosis.

Answer: We used proximal tubular cells from isolations of renal tissue from four different patients. The PTC were isolated after nephrectomies from tissue that was not involved in renal cell carcinoma. However, for data protection reasons, we do not have any age (or other) information on the patients.

It is essential to describe the physiological state of the cells after modeling the inflammatory effect, compare the amounts of production of EVs and their size distribution between cultures before and after the modeling of inflammation.

Answer: That is a very good objection and suggestion. We have now added a sentence on the cell morphology after stimulation. We have also improved Fig. 1 by adding a representative image of an Nanotracking (NTA) measurement of EVs isolated after inflammatory stimulation. We also added the evaluation of the mean EV size and concentration (n=3 vs. 3) and inserted the mean values in the figure.

The research article identified various miRNAs through sequencing. However, to enhance the understanding of the molecular mechanisms of inflammation in PTC, a differential analysis of miRNA expression using the PCR method should be performed.

Answer: This suggestion is also absolutely understandable for us. We have qPCR measurements of miRNAs 146, 147 and 155 from stimulated PTC and controls, done with the commercially available miRCURY system from Qiagen.

These measurements also show an upregulation of all 3 miRNAs after stimulation (by a lower CT value). We have now added a new figure (now Fig. 6) showing these data, but we used the exogeneous spike-in UniSP6 to normalize these data (we have no additional endogeneous housekeeper). In principle, it would of course also be possible to show only the pure CT values without normalization (then in a supplement), in case the reviewer thinks that would make more sense. In addition, we currently have only 3 measurements from cells, and no additional data from EVs (due to the lack of EV samples).

Round 2

Reviewer 2 Report

The authors have successfully addressed all of the reviewer's inquiries and have provided additional experiments in the revised manuscript to further validate the obtained results. Consequently, the reviewer has no further inquiries remaining.